# Research on emission reduction investment strategies for low carbon technology enterprises

Jing Li[1], Tianchen Yang[2]*, Lihua Shi[3]

**1** Shanghai Ocean University, Shanghai, China, **2** Shanghai University of Engineering Science, Shanghai, China, **3** Shanghai University of Engineering Science, Shanghai, China

* m330123405@sues.edu.cn

## Abstract

This study investigates optimal capital allocation strategies for corporate decarbonization initiatives under technological transition constraints. A decision analysis framework grounded in real options theory is developed, incorporating fuzzy-set parameters to address implementation uncertainties in emission mitigation systems. Analytical results demonstrate inverse correlations between critical intervention factors (technological decarbonization efficiency, environmental taxation levels, eco-product market premiums, and fiscal incentive mechanisms) and capital deployment thresholds. Improved technical specifications, reinforced regulatory constraints, positive consumer responses, and targeted subsidy mechanisms synergistically facilitate sustainable infrastructure investments. Comparative evaluations confirm the proposed fuzzy option model's superiority over conventional NPV methods in valuing managerial flexibility and mitigating valuation biases. Sequential option analysis reveals that modular implementation approaches can generate incremental value through adaptive capacity in operational execution. Empirical validation through ''industrial case studies illustrate the framework's practical efficacy in assessing sustainable technology portfolios, offering actionable insights for strategic planning in carbon-intensive industries. This research contributes methodological advancements for timing optimization and risk assessment in environmental technology adoption scenarios.

## Introduction

Against the backdrop of intensifying global climate change, the low-carbon economy has emerged as a core pathway for the international community to achieve sustainable development. The Intergovernmental Panel on Climate Change (IPCC) warns that surpassing a 1.5°C global temperature rise would trigger irreversible ecological catastrophes. Consequently, nations worldwide have proposed carbon neutrality targets and are leveraging policy instruments such as carbon trading and carbon taxes to drive corporate low-carbon transitions. Low-carbon technology enterprises, as

**Data availability statement:** All relevant data are within the paper and its Supporting information files.

**Funding:** This work was funded by Key Laboratory of Information Network Security of Ministry of Public Security (grant number: C23600-05).

**Competing interests:** The authors declare no conflicts of interest.

key players in technological innovation, bear the critical responsibility of developing low-carbon technologies and reducing production-related emissions. However, balancing the economic and environmental benefits of emission reduction investments amid complex policy environments and market volatility remains a pressing challenge for these enterprises [1].

The rapid advancement of digital intelligence technologies has injected new momentum into low-carbon transitions. The integration of the Internet of Things (IoT), big data, and artificial intelligence (AI) enables real-time carbon emission monitoring, optimized energy efficiency, and precise carbon price forecasting. For instance, machine learning-based carbon price prediction models assist enterprises in dynamically adjusting emission reduction strategies [2], while blockchain technology enhances transparency and credibility in carbon trading markets [3]. Nevertheless, digital tools have not fully eliminated uncertainties in investment decision-making. Stochastic fluctuations in carbon prices, dynamic adjustments to government subsidies, and nonlinear shifts in consumer preferences for low-carbon products pose significant challenges to long-term corporate investment planning.

Emission reduction investments by low-carbon technology enterprises face multifaceted complexities. First, carbon trading prices, influenced by energy supply-demand dynamics and geopolitical factors, exhibit characteristics of geometric Brownian motion, making it difficult for firms to accurately assess emission reduction returns [4]. Second, uncertainties in carbon tax rate adjustments and phased changes in government subsidies further compound investment risks. Additionally, the lengthy R&D cycles, high costs, and potential sunk cost risks associated with low-carbon technologies—such as the billion-dollar upfront investments required for Carbon Capture and Storage (CCS) technologies, coupled with operational cost uncertainties due to technological immaturity—amplify these challenges. Traditional Net Present Value (NPV) methods, which overlook managerial flexibility and uncertainty, often undervalue projects, necessitating more adaptive theoretical frameworks for decision-making [5].

Existing research predominantly focuses on single-stage investment optimization under deterministic conditions, with limited exploration of the coupling effects of dynamic policies and market fluctuations. For example, some studies construct investment models based on static carbon price assumptions but fail to account for stochastic carbon price processes. Others incorporate real options theory but neglect the fuzzy nature of policy parameters (e.g., carbon tax rates, subsidy ratios). Furthermore, low-carbon projects typically involve multi-stage processes—R&D, piloting, and commercialization—where investment decisions at each stage are interdependent. Yet, current literature inadequately addresses the modeling of multi-stage compound real options. These theoretical gaps hinder enterprises in formulating strategies that balance short-term returns with long-term flexibility in complex environments.

This paper aims to address three critical questions:

1. How to quantify the impacts of stochastic carbon price fluctuations and policy ambiguities on investment thresholds for low-carbon technology enterprises?

2. How to construct a dynamic decision-making model integrating real options and fuzzy mathematics to accurately evaluate the option value of emission reduction projects?

3. How to optimize the timing and resource allocation of phased investments through a multi-stage compound real options framework?

By addressing these questions, this study seeks to provide low-carbon technology enterprises with theoretically rigorous and practically actionable strategies for emission reduction investments in uncertain environments, thereby supporting the achievement of global "dual carbon" (carbon peaking and carbon neutrality) goals.

## Literature review

The optimization of emission reduction investments in low-carbon technology enterprises under digital intelligence has emerged as a critical area of research, driven by global climate imperatives and technological advancements. Existing studies highlight the interplay between policy instruments, market mechanisms, and technological innovation in shaping corporate strategies. Kumar and Thakur [6] argued that environmental regulations could catalyze innovation, a premise extended to low-carbon technologies by Li et al. [7], who emphasized the role of policy-driven incentives. However, challenges such as path dependency in green technology adoption [8] and high upfront costs of carbon capture and storage (CCS) [9] underscore the need for dynamic decision frameworks.

Real options theory, pioneered by Shao and Sorourkhah [10], provides a robust tool for evaluating irreversible investments under uncertainty. Applications in low-carbon contexts include modeling carbon price volatility via geometric Brownian motion [11] and incorporating policy risks into renewable energy projects [12]. Recent work by Peng et al. [13] integrates real options with carbon trading markets but often neglects the fuzzy nature of policy parameters such as subsidies and tax rates. Fuzzy mathematics, introduced by Wang [14], addresses such ambiguities by quantifying subjective inputs. For instance, trapezoidal fuzzy numbers have been applied to sustainable project evaluations [15], yet their integration with multi-stage low-carbon investments remains limited [16].

Digital intelligence technologies, including artificial intelligence (AI), blockchain, and big data analytics, are reshaping emission reduction strategies. AI-driven carbon price forecasting models enhance decision-making accuracy [17], while blockchain improves transparency in carbon trading [3]. Machine learning algorithms optimize energy efficiency in industrial processes [18], and IoT-enabled monitoring systems reduce operational carbon footprints [19]. Despite these advancements, few studies systematically link digital tools to investment optimization under stochastic and fuzzy conditions [20]. For example, Shah et al. [21] explored AI for carbon market predictions but overlooked policy ambiguity, while Parhamfar and Sadeghkhani [22] combined blockchain with carbon accounting without addressing multi-stage flexibility.

Multi-stage investment strategies, characterized by sequential R&D, piloting, and commercialization phases, require compound real options to capture managerial flexibility. Haegel and Verlinden [23] modeled growth options in R&D, while Ahmed et al. [24] applied compound options to CCS deployment. However, existing frameworks often simplify interactions between policy dynamics and technological risks [25]. Recent advances in fuzzy compound options [26] and stochastic dynamic programming [27] offer promising avenues but lack empirical validation in low-carbon contexts.

Policy instruments such as carbon taxes and subsidies further complicate investment decisions. Hiyagarajan [28] compared price versus quantity mechanisms, while Stoll [29] advocated for adaptive carbon pricing. Empirical studies reveal that subsidies accelerate clean technology adoption [30], yet their effectiveness diminishes under carbon price volatility [31]. The combined effects of digital intelligence, policy ambiguity, and multi-stage risks remain underexplored [32], creating a gap in holistic decision models.

This study addresses three key limitations in prior research: (1) the lack of integration between carbon price stochasticity and fuzzy policy parameters, (2) insufficient application of compound real options to multi-stage low-carbon projects,

and (3) limited use of digital intelligence tools to enhance decision robustness. By bridging these gaps, the proposed framework advances both theoretical and practical understanding of emission reduction investment optimization.

We summarize the most relevant literature in Table 1 and highlight the research gap.

## Problem description and basic assumptions

Under the guidance of government low-carbon policies, analyze the optimization of low-carbon technology investment for low-carbon technology enterprises. Among them, low-carbon technology enterprises are mainly responsible for investment and application of low-carbon technology, and cooperate with universities and research institutions to research and develop core low-carbon technology.

**Assumption 1:** The low-carbon technology enterprise produces only one type of product, with annual sales reaching the enterprise's maximum production capacity $Q$ (a constant). Low-carbon technology deployment preserves production efficiency. Such investments constitute sunk costs with risk-neutral enterprises making these commitments. Upon adopting the low-carbon technology, the enterprise immediately obtains a carbon emission reduction quota, which is maintained stably over the long term. They can be made at time $t = 0$.

**Assumption 2:** Define $e_0$ and $e$ as pre-adoption and post-adoption unit product carbon emissions, respectively. The emission reduction rate ($\eta$) attributable to the low-carbon technology is defined as $\eta = \frac{e_0 - e}{e_0}$

**Assumption 3:** Low-carbon technology deployment entails an upfront capital cost $I$, and the daily operational and maintenance cost is $M$. Technological innovation triggers transitions, leading to a decline in operational costs over time. Assume the daily operational and maintenance cost follows $M = M_0 e^{-lt}$ [33,34].

**Assumption 4:** The carbon emission permit price at time $t$, $p_c(t)$, follows a geometric Brownian motion:

$$d_{p_c(t)} = \mu p_c(t) \, dt + \sigma p_c(t) \, d_{z(t)} \tag{1}$$

where $\mu > 0$ (drift term) and $\sigma > 0$ (volatility) satisfy $0 \leq \mu < r$ ($r$ is the risk-free rate, derived from the capital market's average return), $\sigma$ represents carbon price volatility, and $d_{z(t)}$ is the increment of a standard Wiener process with $d_{z(t)} \sim N(0, d_t)$. Here, $d_t$ denotes an infinitesimally small-time interval.

**Assumption 5:** All else being equal, the enterprise's product price increases after adopting low-carbon technology. According to references [35], potential market demand is represented as 1, assuming a linear function of consumer

**Table 1. Literature comparison.**

| Related Literature | Modeling Approach | Evaluate |
|---|---|---|
| Kumar et al. [6] | Net Present Value (NPV) | Ignores managerial flexibility, irreversibility, and uncertainty beyond discount rates. Static and passive. |
| Li et al. [7] | Basic Real Options (RO) | Assumes precise parameters; struggles with policy ambiguity (fuzzy subsidies/taxes). |
| Wang [14]; Jeevaraj et al. [15] | Fuzzy Models (e.g., Fuzzy NPV) | Limited integration with dynamic investment staging; lacks empirical validation in low-carbon contexts. |
| Haegel et al. [23]; Ahmed et al. [24] | Compound Real Options | Oversimplifies policy-tech interactions; rarely integrates fuzzy or digital tools. |
| Peng et al. [13] | RO and Carbon Trading | Neglects fuzzy policy parameters (e.g., tax/subsidy ambiguity). |
| Shah et al. [21] | AI Forecasting | Focuses on prediction accuracy; ignores embedded options & policy fuzziness. |
| This Paper | Fuzzy Compound Real Options and Digital Enablement | 1. Unifies stochastic carbon prices and fuzzy policy. 2. Values modular, sequential investments (R&D→deployment). 3. Employs digital intelligence (AI/analytics) to handle model complexity. 4. Superiority over NPV: Quantifies flexibility and reduces valuation bias. 5. Empirical Validation: Tested via industrial case studies. |

demand with emission reduction rate ($\eta$), low-carbon preference ($\lambda$), and price per unit product ($P$). Due to the increased demand for low-carbon products ($\lambda$) among consumers, and other conditions being equal, consumers prefer products with lower prices. Therefore, consumer demand is positively correlated with consumer low-carbon preference ($\lambda$) and negatively correlated with retail price ($P$). Therefore, the demand function of consumers is as follows: $q_0 = 1 - P_0$, $q_1 = 1 + \lambda\eta - P_1$, then $q_1 = (1 + \lambda\eta)q_0$, $P_1 = (1+\lambda\eta)P_0$, where $P_0$ is the price per unit product before adopting low-carbon technology. Therefore, the sales premium for low-carbon products is $\omega = \lambda\eta$.

**Assumption 6:** The enterprise is subject to a carbon tax calculated based on its annual total carbon emissions. After adopting low-carbon technology, the reduced carbon tax payment is $F = \eta k e_0 Q$, where $k$ is carbon tax rate.

**Assumption 7:** The government provides a subsidy for the enterprise's low-carbon technology investment, specifically targeting the upfront one-time cost [36,37]. The subsidy ratio is $\theta$.

The main parameters used in this paper are shown in Table 2.

## Model construction and analysis of optimal investment timing

### Model building

Assume that a technology-leading enterprise invests in low-carbon technology at time $t$. Given that this investment yields long-term benefits through a single upfront commitment, the stochastic fluctuations in carbon trading prices create uncertainty risks. Using real options theory, the expected value of carbon emission reductions in the carbon trading market can be evaluated. The present value of benefits gained by the enterprise from adopting low-carbon technology in the carbon trading market is expressed as:

$$V(t) = \int_t^\infty p(s)\, \eta e_0 Q e^{-r(s-t)}\, ds$$

(2)

**Table 2. Parameter list.**

| Parameter | Parameter description |
|---|---|
| $Q$ | The enterprise's maximum production capacity |
| $e_0$ | Carbon emissions per unit product before adoption of low carbon technology |
| $e$ | Carbon emissions per unit product after adoption of low carbon technologies |
| $\eta$ | Emission reduction rate of low carbon technology |
| $I$ | One time investment cost of carbon emission reduction technology |
| $M$ | Daily operation and maintenance cost of carbon emission reduction technology |
| $p_c(t)$ | Carbon emission right price at time $t$ |
| $\mu, \sigma$ | Drift term and variance |
| $r$ | Risk free interest rate |
| $d_{z(t)}$ | Increment of standard Wiener process |
| $\omega$ | Sales premium of low carbon products |
| $P_0$ | Price per unit product before investing in low carbon technology |
| $P_1$ | Price per unit product after investment in low carbon technology |
| $P_T$ | Optimal investment carbon price |
| $\lambda$ | Promotion coefficient of product low-carbon to product market price |
| $\theta$ | Subsidy proportion |
| $\varepsilon$ | Product subsidy coefficient |
| $k$ | Carbon tax rate |

The expected net revenue from low-carbon technology investment comprises five components: carbon credit value (V) in emissions trading markets, tax savings from carbon reduction, product price premium (low-carbon vs. conventional), operational costs, and technology deployment expenses, expressed as:

$$W(t) = E\left[\int_t^\infty \left(p(s)\,\eta e_0 Q + \eta k e_0 Q + \lambda \eta P_0 Q - M\right) e^{-r(s-t)}\,ds - (1-\theta)\,I\right]$$
$$= \frac{\eta e_0 Q p(t)}{r-\mu} + \frac{k\eta e_0 Q + \lambda \eta P_0 Q - M}{r} - (1-\theta)\,I \tag{3}$$

## Threshold carbon price analysis

The risk attitude of a low-carbon technology enterprise is risk-neutral. Hence, it will only think about adopting low-carbon technology when profitability is assured, meaning when the projected investment return of a low-carbon technology enterprise from the adoption of low-carbon technology is greater than zero. Then, Equation (3) must meet the condition of being greater than or equal to 0.That is, when $p(t) \geq \left[\frac{r(1-\theta)I - (k\eta e_0 Q + \lambda \eta P_0 Q - M)}{r\eta e_0 Q}\right](r-\mu)$, it is profitable for the low-carbon technology enterprise to invest in low-carbon technology. Let $p^*$ be the threshold carbon price for the low-carbon technology enterprise to invest in low-carbon technology, then

$$p^* = \left[\frac{r(1-\theta)\,I - (k\eta e_0 Q + \lambda \eta P_0 Q - M)}{r\eta e_0 Q}\right](r-\mu) \tag{4}$$

## The optimal investment timing

Low-carbon technology enterprises possess an investment option prior to investing in Low-carbon technology, with no carbon trading income during the waiting phase. Guided by the goal of maximizing investment returns, they determine when to invest. Specifically, they delay investment until the carbon emission trading price first hits the adoption threshold. This scenario constitutes an optimal stopping problem.

For low-carbon enterprises, pre-investment technology adoption functions as a real option, with valuation tied to emission-reduction option value fluctuations. During deferral periods, these option dynamics follow the Bellman equation under $rWdt = E[dW]$

The call option's value dynamics, derived via Ito's lemma, obey the differential equation:
$\frac{1}{2}\sigma^2 p^2 W'' + (r-\mu) p W' - rW = 0$

The general solution takes the form $W = C_1 p^{\varepsilon_1} + C_2 p^{\varepsilon_2}$, with $\lambda_1$ and $\lambda_2$ as constants.

The conclusion drawn from Dixit and Pingdyck suggests that:

$$C_2 = 0,\ \varepsilon_1 = \frac{1}{2} - \frac{\mu}{\sigma^2} + \sqrt{\left(\frac{\mu}{\sigma^2} - \frac{1}{2}\right)^2 + \frac{2r}{\sigma^2}} > 1$$

Assume Low-carbon tech firms halt waiting and invest at time T to maximize returns. Let be the option value function for Low-carbon tech adoption benefits; then:

$$W_T = \max_{T\geq 0} E\left[\left(\frac{\eta e_0 Q p(t)}{r-\mu} + \frac{k\eta e_0 Q + \lambda \eta P_0 Q - M}{r} - (1-\theta)\,I\right) e^{-r(t-T)}\right] \tag{5}$$

From $E\left[\frac{\eta e_0 Q p(t)}{r-\mu} + \frac{k\eta e_0 Q + \lambda \eta P_0 Q - M}{r} - (1-\theta)\,I\right] e^{-r(t-T)} = \left(\frac{\eta e_0 Q p_T}{r-\mu} + \frac{k\eta e_0 Q + \lambda \eta P_0 Q - M}{r} - (1-\theta)\,I\right)\left(\frac{p(t)}{p_T}\right)^\varepsilon$ and
$\frac{\partial}{\partial p_T}\left(\frac{\eta e_0 Q p_T}{r-\mu} + \frac{k\eta e_0 Q + \lambda \eta P_0 Q - M}{r} - (1-\theta)\,I\right)\left(\frac{p(t)}{p_T}\right)^\varepsilon = 0$, the optimal carbon price for adopting low-carbon technology can be obtained as follows:

$$p_T = \frac{\varepsilon}{\varepsilon - 1}\left[\frac{r(1-\theta)\,I - (k e_0 Q + \lambda \eta P_0 Q - M)}{r\eta e_0 Q}\right](r-\mu) \tag{6}$$

By examining Equations (4) and (6), we see that the optimal adoption carbon price $p_T$ exceeds the threshold carbon price $p^*$ by a factor $\frac{\varepsilon}{\varepsilon-1} > 1$. This shows that $p^*$ marks the break – even point for Low-carbon technology investment, and the ideal carbon price for firms to invest in such technology must be higher.

The supply chain's low-carbon adoption option value (W) under joint optimization is expressed as:

$$W(p(t)) = \begin{cases} \left( \frac{\eta e_0 Q p_T}{r-\mu} + \frac{ke_0 Q + \lambda \eta P_0 Q - M}{r} - (1-\theta)I \right) \left( \frac{p(t)}{p_T} \right)^{\varepsilon}, p(t) < p_T \\ \frac{\eta e_0 Q p_T}{r-\mu} + \frac{ke_0 Q + \lambda \eta P_0 Q - M}{r} - (1-\theta)I, p(t) \geq p_T \end{cases}$$

(7)

The optimal investment timing corresponding to $p_T$ is:

$$T^* = \frac{\ln\left( \frac{p_T}{p_{c0}} \right)}{\mu - \frac{\sigma^2}{2}} = \frac{\ln\left( \frac{\frac{\zeta}{\zeta-1} \left[ \frac{r(1-\theta)I - (ke_0 Q + \lambda \eta P_0 Q - M)}{r\eta e_0 Q} \right](r-\mu)}{p_{c0}} \right)}{\mu - \frac{\sigma^2}{2}},$$

$T^*$ is the first time the carbon price hits the optimal investment price $p_T$

**Corollary 1:** The Low-carbon technology threshold carbon price ($p^*$) and the optimal investment carbon price ($p_T$) are negatively related to the carbon reduction rate ($\eta$). A higher $\eta$ encourages Low-carbon technology firms to invest. Specifically, a greater expected emission reduction after technological upgrades intensifies firms' investment incentives. In this sense, $\eta$ can be viewed as the return on investment for a project. For the same type of emission reduction project, a higher $\eta$ corresponds to a higher return rate and a lower investment threshold.

**Corollary 2:** Both $p^*$ and $p_T$ are negatively correlated with the carbon tax rate ($k$), decreasing as $k$ rises. This indicates that higher values of $k$ mean firms face larger carbon tax bills. This prompts them to adopt Low-carbon technologies sooner to cut emissions and lower their tax costs. Conversely, when $k$ is low, firms may delay investments, awaiting more favorable carbon market returns.

**Corollary 3:** Both $p^*$ and $p_T$ show a negative correlation with the price premium coefficient for low-carbon products ($\lambda$). A higher $\lambda$ lowers $p^*$ and $p_T$, encouraging firms to accelerate investments in low-carbon technologies and produce greener products. Conversely, a lower $\lambda$ raises the investment threshold. In such cases, investing in low-carbon technologies incurs high costs with minimal revenue gains, discouraging early adoption and creating disincentives for investment.

**Corollary 4:** Both $p^*$ and $p_T$ are negatively correlated with the government subsidy ratio ($\theta$). As $\theta$ increases, $p^*$ and $p_T$ decrease, indicating that higher subsidies lower investment barriers and accelerate low-carbon technology adoption.

## Analysis of numerical examples

In order to further explore the impact of various variables on the optimal carbon price using this model, this paper is based on a typical case, where the true values of parameters are substituted into the equation to solve, and Matlab software is used to draw relationship diagrams (Figs 1–4) to intuitively grasp the influence of research variables on investment decisions, making the conclusions of the model clearer and easier to understand.

This section conducted simulation analysis using actual data of CCS technology investment from a low-carbon power technology enterprise. Due to the high level of professional knowledge and huge investment demand required for CCS technology, and the fact that it has not been widely promoted in China, obtaining actual data is relatively difficult. To address this issue, international data, industry analogies, and theoretical estimates were selected for carbon market trading prices and CCS investment costs. Attempt to select project data that is representative of the industry as a reference for policy-making.

## Parameter explanation

(1) Assume that the original emission entity is a thermal power plant. Referring to the case data in the economic analysis part of the China CCS demonstration project of the Asian Development Bank [38], the new investment cost (*I*) for retrofitting the carbon capture, storage and utilization devices is 1.06 billion yuan.

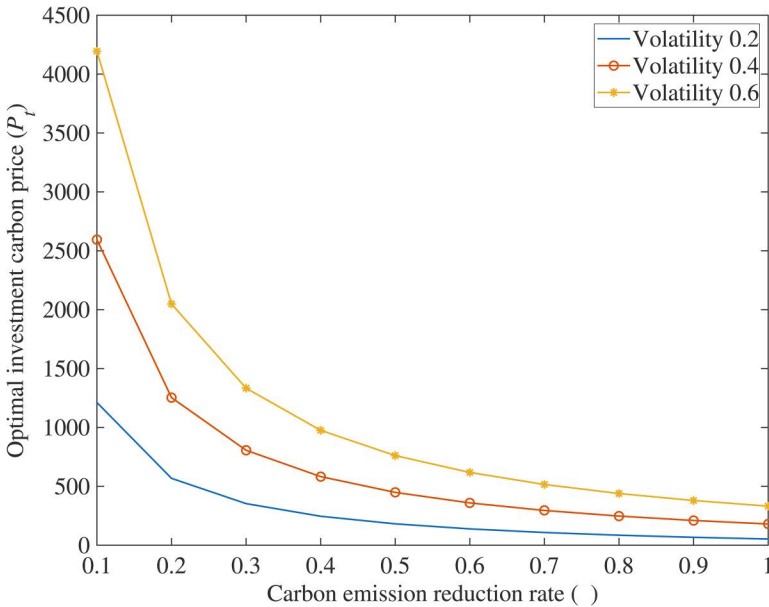

**Fig 1. The impact of carbon emission reduction rate on the on optimal investment carbon price.**

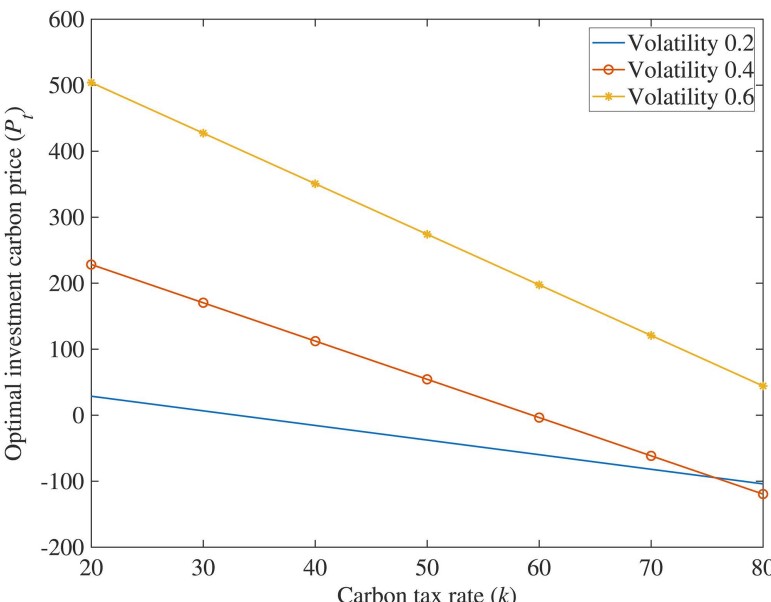

**Fig 2. The impact of carbon tax rate on optimal investment carbon price.**

(2) The annual power generation is the product of the installed capacity and the annual utilization hours (assumed to be 5,700 hours) [39]. That is, the annual power generation $Q$ is 600 MW × 5700 h = 3.42 × 10$^6$ MWh.

(3) Given that the carbon – element content of standard coal is 85%, the molecular weights of C and $CO_2$ are 12 and 44 respectively, and the thermal efficiency is 0.449 [40]. Then the carbon emissions per unit of power generation before

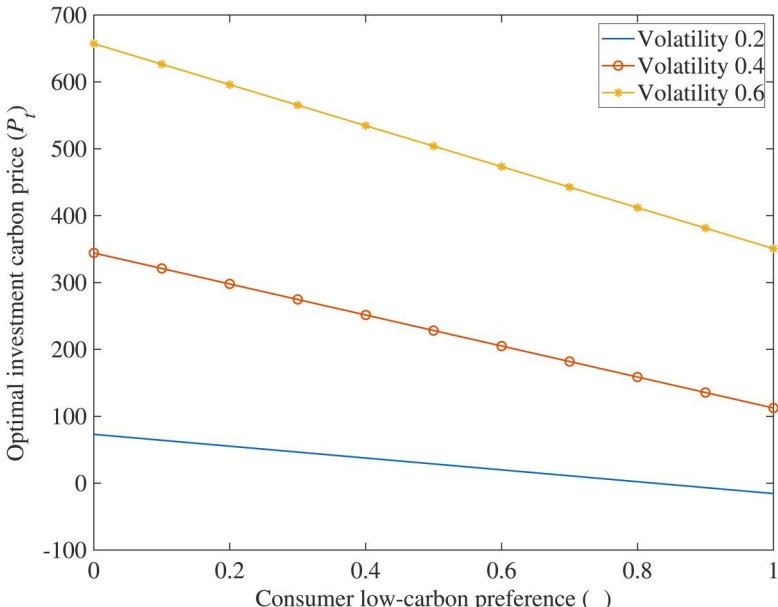

**Fig 3. The impact of consumer low-carbon preference on optimal investment carbon price.**

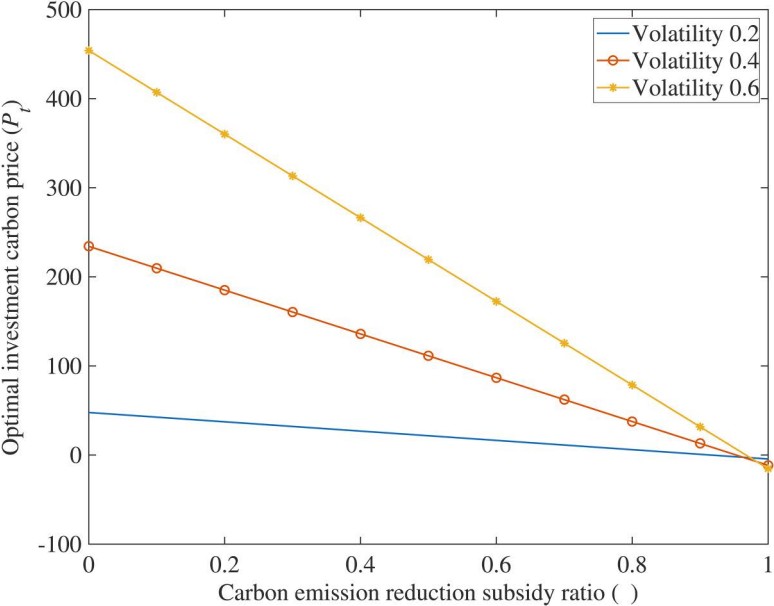

**Fig 4. The impact of carbon emission reduction subsidy ratio on optimal investment carbon price.**

the CCS investment ($e_0$) is $0.449 \times 0.85 \times 44/12 = 1.4$. CCS deployment achieves a 90% carbon capture rate, reducing emissions per unit of power ($e$) generation to $0.449 \times 0.85 \times 44/12 \times (1 - 90\%) = 0.2$.

(4) Based on the tax rate suggestion from the 2010 research group of the Ministry of Ecology and Environment's Planning Institute [41], the initial carbon tax rate ($k$) is set at 20 yuan per ton of $CO_2$.

(5) The risk – free interest rate is set as the 10 – year treasury bond's maturity profit rate at the end of December 2021, which is 2.78% [42]. The coefficient of electricity price ($\lambda$) increase after the CCS investment is 0.05.

(6) Referring to the average on-grid electricity price of 0.31 yuan per kilowatt-hour in 2013 [43], the initial government subsidy ratio ($\theta$) is set at 20%.

(7) After the project is completed and put into use, additional funds are still needed to maintain its operation. And due to the use of carbon dioxide capture and storage technology, it will inevitably lead to higher operating costs than ordinary power plants. Referring to relevant literature [44], assuming that the carbon emission reduction cost of the CCS facility is 450 yuan per ton of $CO_2$, the operating cost of the CCS ($M$) during the entire project period is $450 \times 0.2 \times 3.42 \times 106 = 3.1 \times 10^8$ yuan.

In summary, this study will set the parameters based on the above data as: $I = 1064$, $Q = 1.7$, $e_0 = 1.4$, $e = 0.2$, $k = 20$, $\lambda = 0.05$, $\theta = 0.2$, $M = 31$.

## Sensitivity analysis

Based on the parameter settings in Parameter explanation section. This section uses Matlab software for numerical simulation to analyze the impact of key factors on the optimal investment negotiation price.

(1)  The impact of carbon emission reduction rate ($\eta$) on optimal investment carbon price ($P_T$)

The carbon price volatility is 0.2, 0.4, and 0.6, respectively. Fig 1 depicts the inverse correlation between emission abatement rates and optimal carbon pricing thresholds.

Higher carbon reduction rates lower the optimal carbon price for investment, as increased expected reductions incentivize power generators by enabling greater sale of emission allowances for higher revenue and by delivering significant environmental benefits through reduced emissions, which enhances corporate reputation and increases low-carbon product sales and profits, validating Corollary 1.

(2)  The impact of carbon tax rate ($k$) on optimal investment carbon price ($P_T$)

Fig 2 shows that higher carbon tax rates lower the optimal carbon price adoption, as firms accelerate low-carbon investments to reduce emissions, decrease tax payments, and boost profits, validating Corollary 2. Long-term, carbon taxation effectively cuts $CO_2$ emissions, reduces energy consumption, and shifts energy use patterns, temporarily slowing short-term growth but fostering healthier economic development over the medium to long term.

(3)  The impact of consumer low-carbon preference ($\lambda$) on optimal investment carbon price ($P_T$)

Fig 3 indicates that the optimal adoption carbon price decreases as the price premium coefficient for low-carbon products increases. Specifically, a higher price premium coefficient corresponds to a lower optimal adoption carbon price for supply chain enterprises. This demonstrates that when the price premium coefficient rises, enterprises accelerate investments in low-carbon technologies to reduce carbon emissions. Technology-leading enterprises, aiming to maximize profits during daily operations, benefit from an increased price premium coefficient as it boosts product sales revenue, thereby enhancing corporate profits and thereby bolstering their motivation to invest in low-carbon technologies, validating Corollary 3.

(4)  The impact of carbon emission reduction subsidy ratio ($\theta$) on optimal investment carbon price ($P_T$)

Fig 4 demonstrates that government implementation of cost subsidy policies significantly reduces investment thresholds. The optimal carbon price for investment decreases as the cost subsidy ratio increases, meaning higher subsidy ratios correspond to lower optimal adoption carbon prices for technology-leading enterprises. This indicates that when the government raises the cost subsidy ratio, enterprises expedite adoption of low-carbon technologies to reduce carbon emissions, validating Corollary 4.

## Model expansion

Previous analyses focused on optimizing emission-reduction investments for low-carbon enterprises within a single-stage framework. However, a complete low-carbon technology cycle encompasses multiple phases, such as feasibility studies, project construction, demonstration operations, and commercial operations, each facing distinct uncertainties that yield different option values. Among them, the early stage (feasibility study/construction) mainly includes investment and delay options, while the later stage (demonstration/operation) provides expansion, contraction, and abandonment options. Low-carbon technology investments exhibit intertemporal decision dependency. Early-phase commitments constrain subsequent choices, necessitating decisions on project continuation or scale adjustment; meanwhile, later-phase values influence earlier-phase decisions. At each implementation stage, investors may hold one or multiple options corresponding to strategic flexibility in enterprise operations. Compound real options represent a specialized category where the underlying asset is not a physical asset but another option. Multiple interacting options across time and space form option portfolios. Low-carbon technology projects inherently embody these multi-stage compound real option characteristics.

Under real options theory, such projects derive value from both discounted cash flows during execution and operational flexibility. Thus, total valuation combines net present value (NPV) with embedded strategic option premiums.

Project present value is obtained by discounting expected future cash flows. Discounted cash flow (DCF) methods typically apply when returns are deterministic or predictable. However, future returns in phased low-carbon projects are often non-deterministic and difficult to estimate precisely. This study employs fuzzification to model phase-specific cash flows. Consequently, project values manifest as fuzzy intervals, enhancing the realism of results.

### Net present value of low carbon technology investment by low carbon technology enterprises

Overall, the purpose of low-carbon technology investment is to maximize the investment return. The investment return is composed of two aspects: DACF (Discounted Annual Cash Flow) over the investment horizon versus upfront capital outlay. Low-carbon investments face multiple uncertainties, necessitating probabilistic modeling of projected returns as follows:

$$V = E\left[\sum_{t=1}^{T} \pi_t (1+r)^{-t} - (1-\theta) I\right]$$

(8)

Among them, $\pi_t$ represents the cash flow for year $t$, $r$ represents the discount rate, and $I$ represents the initial investment in low-carbon technology, $\theta$ is the government subsidy ratio.

$\pi_t$ is mainly composed of two parts, namely cash inflows and cash outflows. Cash inflows include selling excess carbon emission rights on the carbon trading market for profit after adopting low-carbon technologies, as well as revenue from the sale of corporate clean products; Cash outflows include equipment operation and maintenance costs. Therefore, the cash flow of low-carbon technology investment by low-carbon technology enterprises can be expressed as:

$$\pi_t = \eta e_0 Q p_c + (P_0 + \omega) Q - M$$

(9)

The net present value (NPV) of low-carbon technology investment by low-carbon technology enterprises can be expressed as:

$$NPV = \sum_{t=1}^{T} (\eta e_0 Q p_c + (P_0 + \omega) Q - M)(1+r)^{-t} - (1-\theta) I$$

(10)

## Construction of composite real option model for low carbon technology projects

Based on the characteristics of low-carbon investment decision processes (featuring compound real options), this section first determines the future revenue cash flow ($S$) of the low-carbon technology project, then derives the n -fold compound real option value ($C_n$) of the R&D investment project.

According to the investment process, it can be seen that the low-carbon investment decision-making process obviously has the characteristics of compound real options: When the enterprise invests $K_n$ at time $t_n$ for the commercial production of products, it is equivalent to buying an ordinary European call option $O_1$. This option gives the enterprise the right to carry out the commercial production of products at time $t_n$, and the strike price is the cost $K_n$ in the commercialization stage.

Suppose that S($t$) represents the cash flow of the future income of the low-carbon technology project. Under the risk-neutral probability measure, S($t$) satisfies the following equation:

$$dS(t) = S(t)[(r - \zeta)dt + \sigma dW(t)]$$

Among them, $r$ represents the risk-free interest rate, $\sigma$ represents the volatility, and both are greater than 0, $\zeta$ represents the leakage of the value of the underlying asset, and $W(t)$ is a standard Brownian motion. The solution of the above equation is: $S(t) = S_0 e^{\left(r - \zeta - \frac{\sigma^2}{2}\right)t + \sigma W(t)}$

Then the value of the n -fold compound real option of the research and development investment project can be expressed as: C=2

$$C_n(S_0) = S_0 e^{-\zeta t_n} N(d_1, d_{2,} \mathrm{L}, d_n; G_n) - \sum_{m=1}^{n} k_m e^{-rt_m} N_m(h_1, h_2, \mathrm{L}, h_m; G_m)$$

(11)

Among them, $d_i = \dfrac{\ln \frac{S_0}{S_i^*} + \left(r - \zeta + \frac{\sigma^2}{2}\right)(t_i - t_{i-1})}{\sigma \sqrt{t_i - t_{i-1}}}$, $i$ =1, 2, …, n

$$h_i = d_i - \sigma \sqrt{t_i - t_{i-1}}, \; i = 1, 2, \ldots, n$$

$N_m(h_1, h_2, \cdots, h_m; G_m)$ is the standard normal distribution function of $m$ dimensions, $G_m$ represents the correlation coefficient matrix of $m$-dimensional standard normal random variables, $G_m = (g_{ij})_{m \times m}$, $m$ =1, 2, …, n, and $g_{ii} = 1$, for any $i < j$, $g_{ij} = \sqrt{\frac{t_i}{t_j}}$. The $S_1^*, S_{2,}^* \cdots, S_n^*$ in the above formulas is the reference value for the enterprise's investment decision-making in research and development and commercial production of products. Its determination method is as follows: $S_n^* = K_n$, and $S_i^*$, $i$ =n-1, n-2, …, 1 can be obtained by solving $C_{n-i}(S_i^*, t_i) = K_i$, $i$ =n-1, n-2, …, 1 successively from the end of the project stage backwards.

## Construction of fuzzy composite real option model for low carbon technology projects

In evaluating low-carbon technology projects under uncertainty, trapezoidal fuzzy numbers (TFNs) effectively represent variable dynamics within compound real option pricing models. These projects face extended timelines and significant operational uncertainties, compounded by early-stage information asymmetries that limit comprehensive data collection. Expected cash flows inherently reflect subjective human judgment—an uncertain and fuzzy factor. Current market conditions, characterized by volatile carbon prices and immature subsidy policies in China, further amplify uncertainties in carbon allowance pricing and government support. Representing cash flows with fuzzy numbers accommodates both real-world human subjectivity and investor management flexibility, yielding more scientifically sound valuations.

This section fuzzifies parameters in the compound real option model to construct a fuzzy pricing framework for low-carbon projects. By expressing investment costs and expected returns at each stage as fuzzy numbers, we establish an n-stage fuzzy compound real option pricing model for low-carbon technology valuation.

 

$$\widetilde{C}_n = e^{-\zeta t_n}\widetilde{S}_0 \otimes \widetilde{N}_n\left(\widetilde{d}_1, \widetilde{d}_2, \cdots, \widetilde{d}_n; G_n\right) \ominus \sum_{m=1}^{n} e^{-rt_m} \otimes \widetilde{k}_m \otimes \widetilde{N}_m\left(\widetilde{h}_1, \widetilde{h}_2, \cdots, \widetilde{h}_m; G_m\right) \tag{12}$$

Among them, $\widetilde{d}_i = \dfrac{\ln\frac{\widetilde{S}_0}{S_i^*} \oplus \left(r - \zeta + \frac{\sigma^2}{2}\right)(t_i - t_{i-1})}{\sigma\sqrt{t_i - t_{i-1}}}$, $i =$ 1, 2, ..., n

$$\widetilde{h}_i = \widetilde{d}_i - \sigma\sqrt{t_i - t_{i-1}}, \ i = 1, 2, \ldots, n$$

The decision-making reference value $S_n^* = E\left(\widetilde{K}_n\right)$ of the discounted present value of the project income at each decision-making time point, and $S_i^*$, where $i = n-1, n-2, \ldots, 1$, can be obtained by solving $C_{n-i}\left(E\left(\widetilde{S}_i\right), t_i\right) = E\left(\widetilde{K}_i\right)$, with $i = n-1, n-2, \ldots, 1$ successively from the end of the project stage backwards.

The *n*-fold real option pricing model does not deny the traditional real option pricing model. Instead, on the basis of the traditional model, it further takes into account the limitation that the subjective prediction of the project's expected cash flow by humans is relatively strong. The value interval range calculated by the *n*-fold fuzzy compound real option pricing model, theoretically speaking, can better show the true value of the low-carbon technology project.

Under the *n*-fold fuzzy compound real option pricing model, the left endpoint $\widetilde{C}_{n,\alpha}^L$ and the right endpoint $\widetilde{C}_{n,\alpha}^R$ of the $\alpha-$ level set interval $\left[\widetilde{C}_{n,\alpha}^L, \widetilde{C}_{n,\alpha}^R\right]$ of the option value $C_n$ of the low-carbon technology project are respectively:

$$\widetilde{C}_{n,\alpha}^L = \widetilde{S}_0^L e^{-\zeta t_n} N\left(\widetilde{d}_{1,\alpha}^L, \widetilde{d}_{2,\alpha}^L, \cdots, \widetilde{d}_{n,\alpha}^L; G_n\right) - \sum_{m=1}^{n} \widetilde{k}_{m,\alpha}^R e^{-rt_m} N_m\left(\widetilde{h}_{1,\alpha}^R, \widetilde{h}_{2,\alpha}^R, \cdots, \widetilde{h}_{m,\alpha}^R; G_m\right) \tag{13}$$

$$\widetilde{C}_{n,\alpha}^R = \widetilde{S}_0^R e^{-\zeta t_n} N\left(\widetilde{d}_{1,\alpha}^R, \widetilde{d}_{2,\alpha}^R, \cdots, \widetilde{d}_{n,\alpha}^R; G_n\right) - \sum_{m=1}^{n} \widetilde{k}_{m,\alpha}^L e^{-rt_m} N_m\left(\widetilde{h}_{1,\alpha}^L, \widetilde{h}_{2,\alpha}^L, \cdots, \widetilde{h}_{m,\alpha}^L; G_m\right) \tag{14}$$

Among them, $\widetilde{d}_{i,\alpha}^L = \dfrac{\ln\frac{\widetilde{S}_{0,\alpha}^L}{S_i^*} \oplus \left(r - \zeta + \frac{\sigma^2}{2}\right)(t_i - t_{i-1})}{\sigma\sqrt{t_i - t_{i-1}}}$, $\widetilde{d}_{i,\alpha}^R = \dfrac{\ln\frac{\widetilde{S}_{0,\alpha}^R}{S_i^*} \oplus \left(r - \zeta + \frac{\sigma^2}{2}\right)(t_i - t_{i-1})}{\sigma\sqrt{t_i - t_{i-1}}}$, $\widetilde{h}_{i,\alpha}^L = \widetilde{d}_{i,\alpha}^L - \sigma\sqrt{t_i - t_{i-1}}$ $\widetilde{h}_{i,\alpha}^R = \widetilde{d}_{i,\alpha}^R - \sigma\sqrt{t_i - t_{i-1}}$

Based on the fuzzy compound real option model, the investment decision-making for low-carbon technology projects is as follows: The decision-makers of low-carbon technology enterprises select a relatively high level (such as above 0.9) as the confidence level. If the decision-makers are satisfied with both the left endpoint $\widetilde{C}_{n,\alpha}^L$ and the right endpoint $\widetilde{C}_{n,\alpha}^R$ of the cut-set interval at this level, then the enterprise can consider initiating the investment in this project; otherwise, it will not consider making the investment for the time being; If the previous – stage R&D is completed and the estimated discounted present value of project income exceeds the relevant decision – making reference value, then proceed with investment or commercial operation.

## Example analysis

This paper applies the above model to the evaluation of a representative CCS investment project of a coal-fired power plant. According to the relevant data and introduction of a certain coal-fired power plant, the coal-fired power plant carried out technical transformation on the original equipment and installed a CCS device. The implementation period of this project is from 2019 to 2041, including a construction period of 2 years and an operation period of 20 years, with a total investment of 130 million yuan.

## Net present value method

The construction and operation process of the CCS project can be summarized into three stages: The first stage is the equipment renovation and CCS supporting construction stage, with an investment cost of 117 million yuan. The construction period is represented by $t_1$, and $t_1 = 2$ years, spanning from 2019 to 2021. After the construction is completed, the

second stage is the Phase I trial operation period, with an investment cost of 1.169 million yuan and a duration of one year, represented by $t_2 = 1$. The third stage is the Phase II trial operation period, with an investment cost of 0.131 million yuan and a duration of one year, represented by $t_3 = 1$. The investment situation of each stage of the CCS project is shown in Table 3.

Then the net present value of this project is: $-11700 + 534.0979 + 770.7798 + 7591.2372 = -2803.886$ (in ten thousand yuan)

The project's NPV stands at $-2803.886$ ten thousand yuan, which is a negative number. The conclusion drawn is that this project does not have investment value, and in terms of economic benefits, this project is not feasible for development.

**Compound real option method.** According to the above compound option model, the parameter explanations are as follows:

(1) Option expiration period $t$

In this project, the construction period $t_1 = 2$, the Phase I construction and rectification period $t_1 = 2$, and the Phase II construction and rectification period $t_3 = 1$.

(2) Present value $S_0$ of projected operational cash flows.

Project NPV reflects projected operational cash flows, non-option-adjusted.

$S_1$, the discounted present value of expected cash – flows in the project's Phase I construction and rectification period, is calculated by the net present value method, ignoring the project's option features.

$S_2$, the discounted present value of expected cash – flows in the project's Phase II construction and rectification period, is net – present – value – calculated, ignoring the project's option characteristics.

(3) Investment cost $K$: The cost invested by investors at each stage of the project.

(4) Risk – free interest rate $r$. In real life, the risk – free interest rate refers to the rate of return obtained by removing the influence of various risk factors on the return, that is, the rate of return obtained by investing funds in a risk – free investment project. In the real estate industry, the current national debt interest rate is often used as the risk – free interest rate. Therefore, in this paper, the risk – free interest rate of this project is determined as the 5 – year national debt interest rate of 3.07% issued by the Ministry of Finance of China in 2021.

(5) Value leakage $\zeta$ of the underlying asset: During the construction and operation of the project, there are situations such as interest, taxes, and insurance premiums. Based on the comprehensive situation, the value leakage is set at 0.02.

(6) Volatility $\sigma$: The value is set at 0.35.

Calculated according to the above formula, we get $C_3(S_0) = 7020.6813$.

**Table 3. Investment situation of each stage of the CCS project.**

| Stage | Duration | Investment Amount | Discounted Cash Flow |
|---|---|---|---|
| Equipment Renovation and CCS Supporting Construction Stage | 2 | 11700 | −11700 |
| Phase I Trial Operation Period | 1 | 1169 | 534.0979 |
| Phase II Trial Operation Period | 1 | 131 | 770.7798 |
| Commercial Operation Period | 18 | 0 | 7591.2372 |

The results show that the option value of this project is 70.206813 million yuan, which is a positive value. Therefore, from the perspective of investment value, the reference suggestion given by the real option method is that this project can be invested in and developed.

**Fuzzy real option method.** According to the previous analysis in the thesis, trapezoidal fuzzy numbers are used to deal with the discounted present values of the investment cost and expected return of the CCS project. Suppose the expected investment cost is a trapezoidal fuzzy number $K_i$, and $K_i = (a_i, b_i, \gamma_i, \beta_i)$. The left endpoint of its core value $[a_i, b_i]$ is $a_i = K_i(1 - \psi)$, the right endpoint is $b_i = K_i(1 + \psi)$, and the left width is equal to the right width, and $\gamma_i = \beta_i = \psi K_i$. Suppose the trapezoidal fuzzy number of the present value of the project's expected return is $S_0$, $S_0 = (a_s, b_s, \gamma_s, \beta_s)$. The left endpoint of its core value $[a_i, b_i]$ is $a_s = S_0(1 - \psi)$, the right endpoint is $b_s = S_0(1 + \psi)$, the core value, the left width is equal to the right width, and $\gamma_s = \beta_s = \psi S_0$. In the following calculations, $\psi = 0.05$ is taken, and the situations of each stage of the CCS project are shown in Table 4.

When $\alpha$ takes some other different values, the $\alpha$-level set (in ten thousand yuan) of the fuzzy value $C_3$ of this CCS project is specifically shown in Table 5.

It can be concluded from the above table that both the left and right endpoints of the project's value interval calculated by the triple fuzzy compound real – option pricing model are positive. As the value of $\alpha$ increases, the range of this interval gradually narrows. Moreover, $C_{3,\alpha}^L$ increases steadily with the increase of $\alpha$, while $C_{3,\alpha}^R$ decreases accordingly.

As shown in the table, when a relatively high confidence level is selected, for example, when $\alpha = 0.95$, the option values $C_{3,\alpha}^L$ and $C_{3,\alpha}^R$ of the CCS project are 6,681.6367 and 7,364.9807 respectively. The resulting value interval of the project is [3,545.9788, 4,229.3228], with both endpoints being positive. This interval provides investors with the reference advice that they should invest in this project.

**Table 4. Situations of each stage of the CCS project under the fuzzy real option method.**

| Stage | Duration | Investment Amount | Discounted Cash Flow |
|---|---|---|---|
| Equipment Renovation and CCS Supporting Construction Stage | 2 | (11115, 12285, 585, 585) | |
| Phase I Trial Operation Period | 1 | (11105, 1227.45, 58.45, 58.45) | |
| Phase II Trial Operation Period | 1 | (124.45, 137.55, 6.55, 6.55) | |
| Commercial Operation Period | 18 | | (7211.6753, 7970.7991, 379.56, 379.56) |

**Table 5. Fuzzy values under different values of α.**

| $\alpha$ | $\left[ C_{3,\alpha}^L, C_{3,\alpha}^R \right]$ |
|---|---|
| 0.8 | [6680.0323, 7366.2955] |
| 0.85 | [6680.5671, 7366.8572] |
| 0.9 | [6681.1019, 7365.4190] |
| 0.91 | [6681.2088, 7365.3313] |
| 0.92 | [6681.3158, 7365.2437] |
| 0.93 | [6681.4228, 7365.1560] |
| 0.94 | [6681.5297, 7365.0684] |
| 0.95 | [6681.6367, 7365.9807] |
| 0.96 | [6681.7437, 7365.8931] |
| 0.97 | [6681.8506, 7365.8054] |
| 0.98 | [6681.9576, 7365.7178] |
| 0.99 | [6681.0645, 7365.6301] |
| 1 | [6681.1715, 7365.5425] |

## Conclusion

### Main finding

This study develops a dynamic investment model combining real options and fuzzy mathematics to overcome traditional NPV limitations, incorporating stochastic carbon prices (modeled via geometric Brownian motion), trapezoidal fuzzy numbers for uncertain parameters (e.g., carbon taxes, subsidies), and multi-stage compound options. Simulations of CCS projects demonstrate how phased investments and operational flexibility enhance valuations. Main finding:

(1)  Carbon Reduction Incentives and Policy Synergy. The study reveals that higher carbon reduction rates ($\eta$), stringent carbon taxes ($k$), strong consumer low-carbon preference ($\lambda$), and increased government subsidies ($\theta$) significantly lower investment thresholds for low-carbon technologies. This highlights the need for policymakers to design synergistic frameworks—such as escalating carbon taxes, consumer incentives for green products, and targeted subsidies—to accelerate corporate adoption. Enterprises should prioritize technologies with higher $\eta$ to maximize carbon market revenues while leveraging subsidies and tax relief to mitigate upfront costs, aligning profitability with environmental goals.

(2) Fuzzy Real Options for Uncertainty Management. Traditional NPV methods underestimate flexibility in low-carbon projects, particularly under volatile carbon prices and ambiguous policy parameters. The proposed fuzzy real options model, incorporating trapezoidal fuzzy numbers, resolves this by quantifying uncertainties and generating interval-based valuations. Managers should adopt such models to evaluate risks tied to subsidy fluctuations, carbon price volatility, and policy shifts. This enables dynamic decision-making, allowing firms to time investments optimally, delay commitments under uncertainty, or pivot strategies as market conditions evolve.

(3) Phased Investments and Operational Flexibility. Multi-stage compound real options analysis demonstrates that breaking investments into sequential phases (e.g., R&D→piloting→scaling) enhances value by embedding managerial flexibility. Case simulations of CCS projects validate that staged strategies allow firms to abandon unviable stages, expand promising ones, or delay commitments until market signals strengthen. Enterprises should adopt modular investment frameworks, allocating resources incrementally while retaining options to adapt to technological advancements, regulatory changes, or shifts in consumer demand. This approach reduces sunk cost risks and ensures long-term resilience in capital-intensive, policy-sensitive sectors.

These insights guide firms in aligning profitability with environmental goals and help policymakers design effective low-carbon transition frameworks.

### Research prospect

Despite its theoretical and practical contributions, this study faces contextual, temporal, and methodological constraints requiring further investigation. Contextually, the model assumes stable regulatory frameworks, potentially limiting efficacy in volatile policy environments (e.g., abrupt subsidy cancellations), while computational complexity hinders real-time application for SMEs without advanced digital infrastructure. Temporally, reliance on static expert estimates for fuzzy parameters risks obsolescence amid rapid technological shifts (e.g., AI-driven CCS efficiency leaps), and the geometric Brownian motion carbon price model may fail to capture "black swan" market disruptions. Methodologically, while the framework outperforms NPV in integrating stochasticity, policy ambiguity, and multi-stage flexibility, it retains dependencies on historical volatility calibration and subjective fuzzy-set boundaries that could introduce bias. Future research should prioritize: (1) developing AI-driven adaptive fuzzy sets for real-time parameter calibration. (2) stress-testing resilience under extreme scenarios (carbon price crashes/subsidy reversals). (3) extending applications to emerging technologies like hydrogen storage and direct air capture.

## Supporting information

**S1 Fig. The impact of carbon emission reduction rate on the on optimal investment carbon price.**
(DOCX)

**S2 Fig. The impact of carbon tax rate on optimal investment carbon price.**
(DOCX)

**S3 Fig. The impact of consumer low-carbon preference on optimal investment carbon price.**
(DOCX)

**S4 Fig. The impact of carbon emission reduction subsidy ratio on optimal investment carbon price.**
(DOCX)

## Author contributions

**Conceptualization:** Jing Li.

**Data curation:** Jing Li.

**Formal analysis:** Jing Li, Lihua Shi.

**Funding acquisition:** Jing Li, Lihua Shi.

**Investigation:** Jing Li.

**Methodology:** Jing Li, Lihua Shi.

**Project administration:** Jing Li, Lihua Shi.

**Resources:** Jing Li, Lihua Shi.

**Software:** Jing Li, Lihua Shi.

**Supervision:** Jing Li, Lihua Shi.

**Validation:** Tianchen Yang.

**Visualization:** Tianchen Yang.

**Writing – original draft:** Tianchen Yang.

**Writing – review & editing:** Tianchen Yang.

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
