## [Decision Letter · Decision Letter 0]

3 Jun 2025

Dear Dr. Yang,

Thank you for submitting your manuscript to PLOS ONE. After careful consideration, we feel that it has merit but does not fully meet PLOS ONE’s publication criteria as it currently stands. Therefore, we invite you to submit a revised version of the manuscript that addresses the points raised during the review process.

We look forward to receiving your revised manuscript.

Kind regards,

Subbarama Kousik Suraparaju

Academic Editor

PLOS ONE

Journal Requirements:

no

4. We note that your Data Availability Statement is currently as follows: All relevant data are included in the manuscript and its supporting information files

5. Please amend your list of authors on the manuscript to ensure that each author is linked to an affiliation. Authors’ affiliations should reflect the institution where the work was done (if authors moved subsequently, you can also list the new affiliation stating “current affiliation:….” as necessary).

6. Please ensure that you refer to Figure 2, 3, and 4, in your text as, if accepted, production will need this reference to link the reader to the figure.

7. We note you have included a table to which you do not refer in the text of your manuscript. Please ensure that you refer to Table 1, 2, and 3, in your text; if accepted, production will need this reference to link the reader to the Table.

Reviewers' comments:

Reviewer's Responses to Questions

**Comments to the Author**

1. Is the manuscript technically sound, and do the data support the conclusions?

Reviewer #1: Yes

Reviewer #2: Yes

2. Has the statistical analysis been performed appropriately and rigorously?

Reviewer #1: Yes

Reviewer #2: N/A

3. Have the authors made all data underlying the findings in their manuscript fully available?

Reviewer #1: Yes

Reviewer #2: Yes

4. Is the manuscript presented in an intelligible fashion and written in standard English?

Reviewer #1: Yes

Reviewer #2: Yes

Reviewer #1: This paper presents a valuable fuzzy real options framework to guide emission reduction investment decisions for SMEs under technological and policy uncertainty. The integration of real options theory, fuzzy mathematics, and compound real options modeling is methodologically sound and improves upon traditional NPV approaches.

However, several points require clarification or improvement:

- The title refers to SMEs, but the CCS case study is more typical of large enterprises. Please clarify the scope or adjust the title accordingly.

- Some sections are mathematically dense and may be hard to follow for broader audiences. A simplified summary or visual diagram of the decision-making process would improve accessibility.

- Consider adding a table to define variables such as η, θ, λ, and ε for clarity.

- The literature review is comprehensive but could benefit from a structured comparison table summarizing key modeling approaches (NPV, real options, fuzzy models, etc.).

- In the simulation setup, it would be helpful to provide more details on parameter values, modeling assumptions, and tools used. So, it can improve transparency to support reproducibility.

- Please ensure all numerical data behind charts or simulations are either included in supplementary files or clearly marked in the manuscript,

Regarding assumptions:

- Assumption 3: Please justify the use of exponential decay in O&M costs with supporting literature or data from CCS or related technologies.

- Assumption 5: The linear link between emission reduction and consumer willingness to pay is likely too simplistic. Consider discussing nonlinear effects influenced by awareness, branding, or segmentation.

- Assumption 7: The focus on a one-time subsidy omits other common forms of support. Please justify this choice—was it due to data constraints, regional policy relevance, or model simplicity?

Reviewer #2: The research developed the study models for Emission Reduction Investment Strategies for Small and Medium-sized Enterprises using Low-Carbon Technologies. The research evaluated the critical factors that simultaneously supported sustainable infrastructure investments. The evaluation is compared with the conventional method to show the superiority of the proposed plan. In addition, the proposed model was implemented on some case studies to verify its practical effectiveness. However, we have included some comments to enhance the quality of the manuscript:

1. As the author explains the weaknesses of several conventional methods, the author needs to explain several fundamental weaknesses of the proposed plan in terms of context of use, time, and environment. Besides, the authors can explore more about the advantages and disadvantages of the proposed method

2. All figures still need to be improved to make it easier to observe and read.

3. The paper is too long. The writing can be made shorter, more concise, and clearer.

4. Further, the author can address some potential detailed future research issues in the conclusion section.

**Do you want your identity to be public for this peer review?** For information about this choice, including consent withdrawal, please see our Privacy Policy

Reviewer #1: No

Reviewer #2: No

---

## [Author Response · Author response to Decision Letter 1]

24 Aug 2025

To Reviewer #1:

1. The title refers to SMEs, but the CCS case study is more typical of large enterprises. Please clarify the scope or adjust the title accordingly.

Thank you for your suggestion, it is very helpful for us to improve the manuscript. We will adjust the title of the paper to “Research on Emission Reduction Investment Strategies for Low Carbon Technology Enterprises” to match the title of the pape.

Changes Made: According to this comment, the changes have been highlighted in blue in the title section of the paper.

2. Some sections are mathematically dense and may be hard to follow for broader audiences. A simplified summary or visual diagram of the decision-making process would improve accessibility.

Thank you for your suggestions, it is very helpful for us to improve the manuscript. We have added a simplified summary in the mathematically intensive section to improve accessibility.

Changes Made: According to this comment, the changes have been highlighted in blue in the Construction of composite real option model for low carbon technology projects and Construction of fuzzy composite real option model for low carbon technology projects sections of the manuscript.

3. Consider adding a table to define variables such as η, θ, λ, and ε for clarity.

Thank you for your suggestion, it is very helpful for us to improve the manuscript. We have added a table to define variables.

Changes Made: According to this comment, the changes have been highlighted in blue in the Problem description and basic assumptions section.

4. The literature review is comprehensive but could benefit from a structured comparison table summarizing key modeling approaches (NPV, real options, fuzzy models, etc.).

Thank you for your suggestion, it is very helpful for us to improve the manuscript. We have added a structured comparison table summarizing key modeling methods.

Changes Made: According to this comment, the changes have been highlighted in blue in the Literature review section.

5. In the simulation setup, it would be helpful to provide more details on parameter values, modeling assumptions, and tools used. So, it can improve transparency to support reproducibility.

Thank you very much for your suggestion, it is very helpful for us to improve the manuscript. This method can indeed improve transparency to support reproducibility. Therefore, in the simulation setup, we provide details about parameter values and tool usage.

Changes Made: According to this comment, we have modified and updated the Analysis of numerical examples section and highlighted it in blue.

6. Please ensure all numerical data behind charts or simulations are either included in supplementary files or clearly marked in the manuscript,

Thank you very much for your suggestion, it is very helpful for us to improve the manuscript. We clearly marked all numerical data behind the charts or simulations in the manuscript. And the source code of the simulation Figures is provided in the folder Fig1-Fig4

Changes Made: According to this comment, the changes have been highlighted in blue in the Parameter explanation section.

7. Assumption 3: Please justify the use of exponential decay in O&M costs with supporting literature or data from CCS or related technologies.

Thank you for your suggestions, it is very helpful for us to improve the manuscript. We referred to references [34] and [35] to demonstrate the rationality of using exponential decay in operational costs.

[34] Steffen, Bjarne, et al. Experience curves for operations and maintenance costs of renewable energy technologies. Joule. 2020; 10: 359-375. doi: 10.1016/j.joule.2019.11.012

[35] Gaëtan Frusque, Daniel Mitchell, Jamie Blanche, David Flynn, Olga Fink, Non-contact sensing for anomaly detection in wind turbine blades: A focus-SVDD with complex-valued auto-encoder approach. Mechanical Systems and Signal Processing. 2024; 208: 111022. https://doi.org/10.1016/j.ymssp.2023.111022.

Changes Made: According to this comment, relevant changes are highlighted in blue in the Problem description and basic assumptions section.

8. Assumption 5: The linear link between emission reduction and consumer willingness to pay is likely too simplistic. Consider discussing nonlinear effects influenced by awareness, branding, or segmentation.

Thank you for your suggestions, it is very helpful for us to improve the manuscript. We refer to reference [36] and further characterize the relationship between emissions reduction and consumer willingness to pay, based on consumers' preference for low-carbon.

[36] Yuan H, Zhang L, Cao B B, et al. Optimizing traceability scheme in a fresh product supply chain considering product compe-tition in blockchain era, Expert Syst Appl. 2024; 258: 125127. https://doi.org/10.1016/j.eswa.2024.125127

Changes Made: According to this comment, relevant changes are highlighted in blue in the Problem description and basic assumptions section.

9. Assumption 7: The focus on a one-time subsidy omits other common forms of support. Please justify this choice—was it due to data constraints, regional policy relevance, or model simplicity?

Thank you for your suggestions, it is very helpful for us to improve the manuscript. We referred to reference [37, 38], and in order to simplify the model, this choice is reasonable.

[37] Li, Bin, Timing Decision of Low-Carbon Technology Investment Adoption by High Energy Consuming Enterprises under Carbon Trading and Subsidies, Journal of Environmental and Public Health. 2022; 8: 9848994. https://doi.org/10.1155/2022/9848994

[38] Hoda Talebian, Omar E. Herrera, Walter Mérida, Policy effectiveness on emissions and cost reduction for hydrogen supply chains: The case for British Columbia, International Journal of Hydrogen Energy. 2021; 46: 998-1011. https://doi.org/10.1016/j.ijhydene.2020.09.190.

Changes Made: According to this comment, relevant changes are highlighted in blue in the Problem description and basic assumptions section.

To Reviewer #2:

1. As the author explains the weaknesses of several conventional methods, the author needs to explain several fundamental weaknesses of the proposed plan in terms of context of use, time, and environment. Besides, the authors can explore more about the advantages and disadvantages of the proposed method

Thank you for your suggestions, it is very helpful for us to improve the manuscript. We explained several basic drawbacks of the proposed plan from the perspectives of methodology, time, and environment. In addition, we can further explore the advantages and disadvantages of the proposed method.

Changes Made: According to this comment, the relevant changes are highlighted in blue in the Research Prospects section.

2. All figures still need to be improved to make it easier to observe and read.

Thank you for your suggestions, it is very helpful for us to improve the manuscript. We further optimized all the figures to make them easier to observe and read.

Changes Made: According to this comment, corresponding changes have been highlighted in blue in the Sensitivity analysis section.

3. The paper is too long. The writing can be made shorter, more concise, and clearer.

Thank you for your suggestions, it is very helpful for us to improve the manuscript. We further polished the paper, making it shorter, more concise, and clearer.

Changes Made: According to this comment, corresponding changes have been highlighted in blue in the Model construction and analysis of optimal investment timing, Analysis of numerical examples, Model expansion and Conclusion sections of the manuscript.

4. Further, the author can address some potential detailed future research issues in the conclusion section.

Thank you for your suggestions, it is very helpful for us to improve the manuscript. We have proposed some potential detailed future research questions in the conclusion section.

Changes Made: According to this comment, corresponding changes have been highlighted in blue in the Conclusion section.

---

## [Decision Letter · Decision Letter 1]

7 Sep 2025

Dear Dr. Yang,

Thank you for submitting your manuscript to PLOS ONE. After careful consideration, we feel that it has merit but does not fully meet PLOS ONE’s publication criteria as it currently stands. Therefore, we invite you to submit a revised version of the manuscript that addresses the points raised during the review process.

We look forward to receiving your revised manuscript.

Kind regards,

Subbarama Kousik Suraparaju

Academic Editor

PLOS ONE

Journal Requirements:

Reviewers' comments:

Reviewer's Responses to Questions

**Comments to the Author**

Reviewer #1: All comments have been addressed

Reviewer #2: All comments have been addressed

2. Is the manuscript technically sound, and do the data support the conclusions?

Reviewer #1: Yes

Reviewer #2: Yes

3. Has the statistical analysis been performed appropriately and rigorously?

Reviewer #1: Yes

Reviewer #2: N/A

4. Have the authors made all data underlying the findings in their manuscript fully available?

Reviewer #1: Yes

Reviewer #2: Yes

5. Is the manuscript presented in an intelligible fashion and written in standard English?

Reviewer #1: Yes

Reviewer #2: Yes

Reviewer #1: (No Response)

Reviewer #2: All comments have been addressed, particularly related to some potential detailed future research issues.

**Do you want your identity to be public for this peer review?** For information about this choice, including consent withdrawal, please see our Privacy Policy

Reviewer #1: No

Reviewer #2: No

---

## [Author Response · Author response to Decision Letter 2]

1 Oct 2025

We sincerely thank the editors and reviewers for taking time out of their busy schedules to review our manuscripts. Thanks to the editors and all reviewers for their valuable comments, which we use to improve the quality of our manuscripts. The editor's and reviewer's comments were crucial to us, and we have carefully considered and incorporated them into the revision of the manuscript. In order to clearly show our response and changes to the manuscript, we have used the following format: reviewers' comments are listed in italics and numbered as necessary, our response is in regular font, and changes and additions to the manuscript are highlighted in blue. Once again, I would like to thank the editors and reviewers for their excellent guidance and support.

Comments marked in the manuscript by the editors and reviewers

Reviewer:

1. If the authors have adequately addressed your comments raised in a previous round of review and you feel that this manuscript is now acceptable for publication, you may indicate that here to bypass the “Comments to the Author” section, enter your conflict of interest statement in the “Confidential to Editor” section, and submit your "Accept" recommendation.

Reviewer #1: All comments have been addressed

Reviewer #2: All comments have been addressed

Thank you for your affirmation. We will continue to improve the manuscript.

2. Is the manuscript technically sound, and do the data support the conclusions?

Reviewer #1: Yes

Reviewer #2: Yes

Thank you for your affirmation. We will continue to improve the manuscript.

3. Has the statistical analysis been performed appropriately and rigorously?

Reviewer #1: Yes

Thank you for your affirmation. We will continue to improve the manuscript.

Reviewer #2: N/A

Thank you for your suggestion, it is very helpful for us to improve the manuscript. We have cited specific literature to set the parameters of the paper more reasonably, ensuring that the statistical analysis is more appropriate and reasonable.

Changes Made: According to this comment, the changes have been highlighted in blue in the Parameter explanation section of the paper.

4. Have the authors made all data underlying the findings in their manuscript fully available?

Reviewer #1: Yes

Reviewer #2: Yes

Thank you for your affirmation. We will continue to improve the manuscript.

5. Is the manuscript presented in an intelligible fashion and written in standard English?

Reviewer #1: Yes

Reviewer #2: Yes

Thank you for your affirmation. We will continue to improve the manuscript.

6. Review Comments to the Author

Reviewer #1: (No Response)

Reviewer #2: All comments have been addressed, particularly related to some potential detailed future research issues.

Thank you for your affirmation. We will continue to improve the manuscript.

Additional Editor Comments:

Thank you for submitting your manuscript to PLOS ONE. After careful consideration, we feel that it has merit but does not fully meet PLOS ONE’s publication criteria as it currently stands. Therefore, we invite you to submit a revised version of the manuscript that addresses the points raised during the review process.

Thank you for your suggestions. We have further revised the paper based on the comments of the reviewers. In addition, we have modified the reference format of the paper according to the journal's formatting requirements and inserted Figure into the paper instead of displaying it in Supporting Information.

Changes Made: According to this comment, the changes have been highlighted in blue in Parameter explanation, Sensitivity analysis, References and Supporting information sections of the paper.

---

## [Editor Report · Decision Letter 2]

2 Oct 2025

Research on Emission Reduction Investment Strategies for Low Carbon Technology Enterprises

PONE-D-25-22903R2

Dear Dr. Yang,

We’re pleased to inform you that your manuscript has been judged scientifically suitable for publication and will be formally accepted for publication once it meets all outstanding technical requirements.

Kind regards,

Subbarama Kousik Suraparaju

Academic Editor

PLOS ONE
---

## [Editor Report · Acceptance letter]

PONE-D-25-22903R2

PLOS ONE

Dear Dr. Yang,

I'm pleased to inform you that your manuscript has been deemed suitable for publication in PLOS ONE. Congratulations! Your manuscript is now being handed over to our production team.

Kind regards,

on behalf of

Dr. Subbarama Kousik Suraparaju

Academic Editor

PLOS ONE